# Selective blockade of Ca$_v$1.2 (α1C) versus Ca$_v$1.3 (α1D) L-type calcium channels by the black mamba toxin calciseptine

Pietro Mesirca [1,2] ✉, Jean Chemin [1,2,6], Christian Barrère [1,2,6], Eleonora Torre [1,2], Laura Gallot [1,2], Arnaud Monteil [1,2,3], Isabelle Bidaud [1,2], Sylvie Diochot [2,4], Michel Lazdunski [2,4], Tuck Wah Soong [5], Stéphanie Barrère-Lemaire [1,2], Matteo E. Mangoni [1,2] & Joël Nargeot [1,2] ✉

L-type voltage-gated calcium channels are involved in multiple physiological functions. Currently available antagonists do not discriminate between L-type channel isoforms. Importantly, no selective blocker is available to dissect the role of L-type isoforms Ca$_v$1.2 and Ca$_v$1.3 that are concomitantly co-expressed in the heart, neuroendocrine and neuronal cells. Here we show that calciseptine, a snake toxin purified from mamba venom, selectively blocks Ca$_v$1.2-mediated L-type calcium currents (I$_{CaL}$) at concentrations leaving Ca$_v$1.3-mediated I$_{CaL}$ unaffected in both native cardiac myocytes and HEK-293T cells expressing recombinant Ca$_v$1.2 and Ca$_v$1.3 channels. Functionally, calciseptine potently inhibits cardiac contraction without altering the pacemaker activity in sino-atrial node cells, underscoring differential roles of Ca$_v$1.2− and Ca$_v$1.3 in cardiac contractility and automaticity. In summary, calciseptine is a selective L-type Ca$_v$1.2 Ca$^{2+}$ channel blocker and should be a valuable tool to dissect the role of these L-channel isoforms.

By mediating calcium influx in a wide range of cell types, voltage-gated calcium channels (VGCCs) are critically involved in many physiological functions, including contraction in cardiac and smooth muscles, hormone secretion in endocrine cells, synaptic transmission in neurons, gene transcription and regulation of enzyme activity (for review see[1,2]). VGCCs have been classified in three families, based on the pore-forming α$_1$ subunit, i.e. Ca$_v$1, Ca$_v$2 and Ca$_v$3[3]. VGCCs are hetero-oligomeric complexes constituted by the α1-subunits defining the Ca$^{2+}$ channel subtype associated to different ancillary subunits (α2δ, β, and γ). The Ca$_v$1 family is formed by four distinct α$_1$ subunit isoforms (Ca$_v$1.1-Ca$_v$1.4) underlying "long-lasting" (L)-type Ca$^{2+}$ currents. The Ca$_v$2 family includes three α$_1$ isoforms (Ca$_v$2.1-Ca$_v$2.3) encoding "high-voltage activated" Ca$^{2+}$ currents referred to as P/Q-type, N-type and

R-type. The Ca$_v$3 family of α$_1$ subunit isoforms (Ca$_v$3.1 through Ca$_v$3.3) mediates "low-voltage activated" transient (T)-type Ca$^{2+}$ currents and is not associated to ancillary subunits. While Ca$_v$2 channel isoforms are highly expressed in the nervous system[1], cardiomyocytes can express T-type and L-type Ca$^{2+}$ channels underlying I$_{CaT}$ and I$_{CaL}$, respectively[4].

The heart expresses two L-type Ca$^{2+}$ channels isoforms, Ca$_v$1.2 and Ca$_v$1.3[5]. Ca$_v$1.2 is ubiquitously expressed in the heart ventricular and supraventricular chambers, as well as in the sinoatrial (SAN) and atrio-ventricular (AVN) nodes[5]. Ca$_v$1.2 is also the predominant L-type isoform in adult ventricle, where it couples excitation to contraction and con-stitutes the target of the cAMP/PKA mediated positive inotropic effect on cardiac contractility exerted by catecholamines[6,7]. Ca$_v$1.3 is expressed in the atria and is strongly expressed in the SAN and AVN. Ca$_v$1.3 plays a

¹Institut de Génomique Fonctionnelle, Université de Montpellier, CNRS, INSERM, 34094 Montpellier, France. ²Laboratory of Excellence Ion Channels, Science & Therapeutics, F-06560 Valbonne, France. ³Department of Physiology, Faculty of Medicine Siriraj Hospital, Mahidol University, Bangkok, Thailand. ⁴Uni-versité Côte d'Azur, CNRS, IPMC (Institut de Pharmacologie Moléculaire et Cellulaire), FHU InovPain (Fédération Hospitalo-Universitaire "Innovative Solutions in Refractory Chronic Pain"), F-06560 Valbonne, France. ⁵Department of Physiology, Yong Loo Lin School of Medicine, National University of Singapore, Singapore, Singapore. ⁶These authors contributed equally: Jean Chemin, Christian Barrère. ✉e-mail: pietro.mesirca@igf.cnrs.fr; joel.nargeot@igf.cnrs.fr

major role in SAN pacemaker activity and impulse conduction through the AVN. In this respect, mice in which $Ca_v1.3$ has been genetically ablated ($Ca_v1.3^{-/-}$) show prominent SAN bradycardia, susceptibility to atrial fibrillation and atrioventricular blocks[8, 9]. In SAN pacemaker myocytes, $Ca_v1.3$-mediated $I_{CaL}$ ($I_{Cav1.3}$) is characterized by a more negative activation threshold and slower inactivation kinetics than $Ca_v1.2$-mediated $I_{CaL}$ ($I_{Cav1.2}$) and is essential for the generation of diastolic depolarization underlying SAN automaticity[8, 10] (for review see[11]).

Despite the importance of dissecting the physiopathological roles of $Ca_v1.2$ and $Ca_v1.3$ channels in cardiac myocytes and neuronal cells, no selective pharmacological tool is currently available. Both $Ca_v1.2$ and $Ca_v1.3$ are sensitive to antagonist and agonist dihydropyridines (DHPs) and to non-DHP drugs, such as phenylalkylamine (i.e. Verapamil) and benzothiazepine-like compounds (i.e. Diltiazem) without significant selectivity[1].

Animal venoms constitute an immense reservoir for identification of new toxins and toxin-derived tools for modulating the physiological function of ion channels. In particular, toxins showing three-fingers molecular scaffold class include several cardiotoxic peptides that abound in snake venoms. Three fingers toxin peptides potently block a variety of proteins including nicotinic receptors[12], acetylcholinesterase[13], acid-sensitive ion channels (ASIC)[14], as well as cytotoxins that disrupt the lipid membrane[15]. More than 30 years ago, M. Lazdunski 's group purified a 60-amino acid three-fingers polypeptide from the venom of the black mamba (*Dendroaspis polylepis*) called calciseptine (Cas). Cas was shown to selectively block $I_{CaL}$, to abolish all contraction in portal vein, thoracic aorta and uterine smooth muscle and to completely eliminate cardiac contractions[16]. These properties constitute clear adaptation to snake's predatory behavior to induce very rapid and robust hypotension and dramatic negative inotropic effect in the prey, also in combination with the effects of the other venom components. Interestingly, Cas specifically blocks $I_{CaL}$, without affecting N-type, T-type and also the skeletal muscle ($Ca_v1.1$) L-type $Ca^{2+}$ channels isoforms[16]. In addition, previous data show that Cas does not affect voltage sensitive $Na^+$ and $K^+$ channels in cardiac ventricular cells, NlE115 neuroblastoma cells, RINmSF and HIT insulinomas and chicken dorsal root ganglia[16].

In this work, we show that Cas selectively blocks $Ca_v1.2$ channels, while $Ca_v1.3$ is resistant to Cas. We show that Cas potently decreases cardiac contractility without affecting heart rate of Langendorff-perfused mouse heart. In line with potent inhibition of contraction, Cas blocks $I_{Cav1.2}$ in mouse ventricular myocytes. We show that while Cas partially inhibits total $I_{CaL}$ ($I_{Cav1.2}+I_{Cav1.3}$) in SAN myocytes from wild-type mice, it totally blocks $I_{CaL}$ in myocytes from $Ca_v1.3^{-/-}$ counterparts, where only $Ca_v1.2$ is functional. Selectivity of Cas blockade of $Ca_v1.2$ versus $Ca_v1.3$ is confirmed in HEK-293T expressing recombinant $Ca_v1.2$, $Ca_v1.3$ channels. Notably, the currents mediated by both $Ca_v1.3_{42a}$ and $Ca_v1.3_{42}$ splice variants—characterized by short and long C-terminus, respectively[17]- are totally resistant to Cas. Finally, while DHP nifedipine robustly slows pacemaker activity of SAN myocytes, selective blockade of $I_{Cav1.2}$ by Cas reduces the action potential amplitude, leaving unaffected the cell firing rate. In conclusion, our study shows that Cas selectively blocks $Ca_v1.2$ channels in the heart, providing an important tool in the panel of calcium channel modulators to help dissect the role of this channel isoform in heart physiology and automaticity.

## Results

### Calciseptine inhibited contraction and spared automaticity of Langendorff perfused hearts
We studied the effects of Cas on cardiac contractility and automaticity in isolated hearts from mice before and after perfusion of native toxin (100 nM). Perfusion of Cas strongly reduced left ventricular (LV) contraction amplitude ($51 \pm 3$ mmHg vs $9 \pm 2$ mmHg. Fig. 1a, b), without changing baseline diastolic pressure ($8 \pm 2$ mmHg vs $8 \pm 3$ mmHg. Fig. 1a, b). Averaged inter-beat (RR) intervals were not affected by toxin

perfusion ($273 \pm 17$ ms vs $271 \pm 17$ ms, $p > 0.05$. Fig. 1c), indicating that Cas did not disrupt the cardiac pacemaker mechanism (automaticity). In addition, we did not observe episodes of arrhythmia upon perfusion of Cas. We then evaluated the effects of Cas on heart inotropism and lusitropism by measuring the contraction velocity (dP/dt max) and the relaxation velocity (dP/dt min), respectively. Perfusion of Cas (100 nM), significantly reduced dP/dt max ($1.7 \pm 0.2$ mmHg/ms vs $0.4 \pm 0.1$ mmHg/ms) and dP/dt min ($-1.2 \pm 0.1$ mmHg/ms vs $-0.1 \pm 0.1$ mmHg/ms), indicating negative inotropic and lusitropic effects of the toxin (Fig. 1d). Furthermore, relaxation time constant (T) and time-to-peak interval (or time to maximal pressure), parameters related to contraction duration, were not affected by perfusion of Cas ($10 \pm 1$ ms vs $13 \pm 2$ ms, and $54 \pm 1$ ms vs $50 \pm 3$ ms, respectively. Supplementary Fig. 1). Taken together these data demonstrated differential effect of Cas on heart contractility and automaticity, two distinct functions that are reliant on L-type voltage dependent $Ca^{2+}$ channels subtypes expression and regulation.

### Calciseptine potently and selectively blocked L-type $Ca^{2+}$ current ($I_{Cav1.2}$) in isolated ventricular myocytes and $I_{Cav1.2}$ in HEK-293T cells expressing $Ca_v1.2$
We evaluated the effects of Cas on isolated adult mouse ventricular myocytes in which $I_{CaL}$ is mediated by $Ca_v1.2$ ($I_{Cav1.2}$)[8]. For voltages ranging from $-25$ mV to $+40$ mV, $I_{Cav1.2}$ in rod-shape ventricular myocytes was dose-dependently reduced by perfusion of Cas (Fig. 2). Application of nifedipine ($3 \mu M$), blocked residual $Ca^{2+}$ current in myocytes exposed to Cas 100 nM and Cas 300 nM, but failed to have an effect when cells were previously perfused with Cas $1 \mu M$ (Supplementary Fig. 2).

Whole cell recording of $I_{CaL}$ on recombinant $Ca_v1.3_{42a}$ and $Ca_v1.3_{42}$ channels[18] and $Ca_v1.2$ channels expressed in HEK-293T cells before and after perfusion is shown in Fig. 3. Using a one-step voltage clamp protocol (holding potential $-60$ mV, voltage test 0 mV) we showed that Cas exhibited an inhibitory effect on $I_{CaL}$ in $Ca_v1.2$ transfected cells in a dose dependent manner. 100 nM Cas inhibited the 37% of $I_{Cav1.2}$ (Supplementary Fig. 3a), while 300 nM and $1 \mu M$ Cas produced respectively an inhibition of 70% and 80% of $I_{Cav1.2}$ (Fig. 3a). Dose-effect relationship showed that Cas inhibits $I_{Cav1.2}$ with $IC_{50}$ value of $92 \pm 18$ nM (Supplementary Fig. 3b). We then investigated Cas effects on recombinant $Ca_v1.3$ channels. We did not record a statistically significant reduction of recombinant $I_{Cav1.3-42a}$ and $I_{Cav1.3-42}$ expressed in HEK-293T cells by Cas at all concentrations tested (up to $1 \mu M$) (Fig. 3a). Averaged current-to-voltage (I-V) relationships recorded upon perfusion of 300 nM of Cas showed inhibition of $I_{Cav1.2}$ by -60% at membrane voltages ranging from $-20$ mV to $+30$ mV (Fig. 3b, Supplementary Fig. 4). Application of Cas significantly shifted the activation curve of $I_{Cav1.2}$. The $V_{0.5}$ for activation was $-3 \pm 2$ mV in control conditions and $3 \pm 2$ mV during Cas application, respectively ($n = 8$, Wilcoxon test, $p = 0.0078$). We observed no significant effect of Cas on both $I_{Cav1.3-42a}$ and $I_{Cav1.3-42}$ at all voltages tested (Fig. 3b, Supplementary Fig. 4). These data clearly indicate that Cas is a potent inhibitor of $Ca_v1.2$ channels irrespective of the $Ca_v1.3$ channel variant. We next investigated Cas selectivity on recombinant $Ca_v3.1$ T-type, $Ca_v2.2$ N-type and $Ca_v2.1$ P/Q type calcium channels expressed in HEK-293T cells. Moreover, knowing the propensity for several calcium channel blockers to affect human ether-a-go-go-related gene (hERG)[19] outward current, we also evaluated the effect of Cas on recombinant hERG current expressed in *Xenopus* oocytes. Cas had no effect on the aforementioned recombinant currents investigated (Supplementary Fig. 5 and Supplementary Fig. 6).

### Calciseptine selectively inhibited $I_{Cav1.2}$ in isolated mouse SAN myocytes
We studied the effects of Cas perfusion on total $I_{CaL}$ in isolated SAN pacemaker cells from wild-type and mutant $Ca_v1.3^{-/-}$ mice (Fig. 4). We

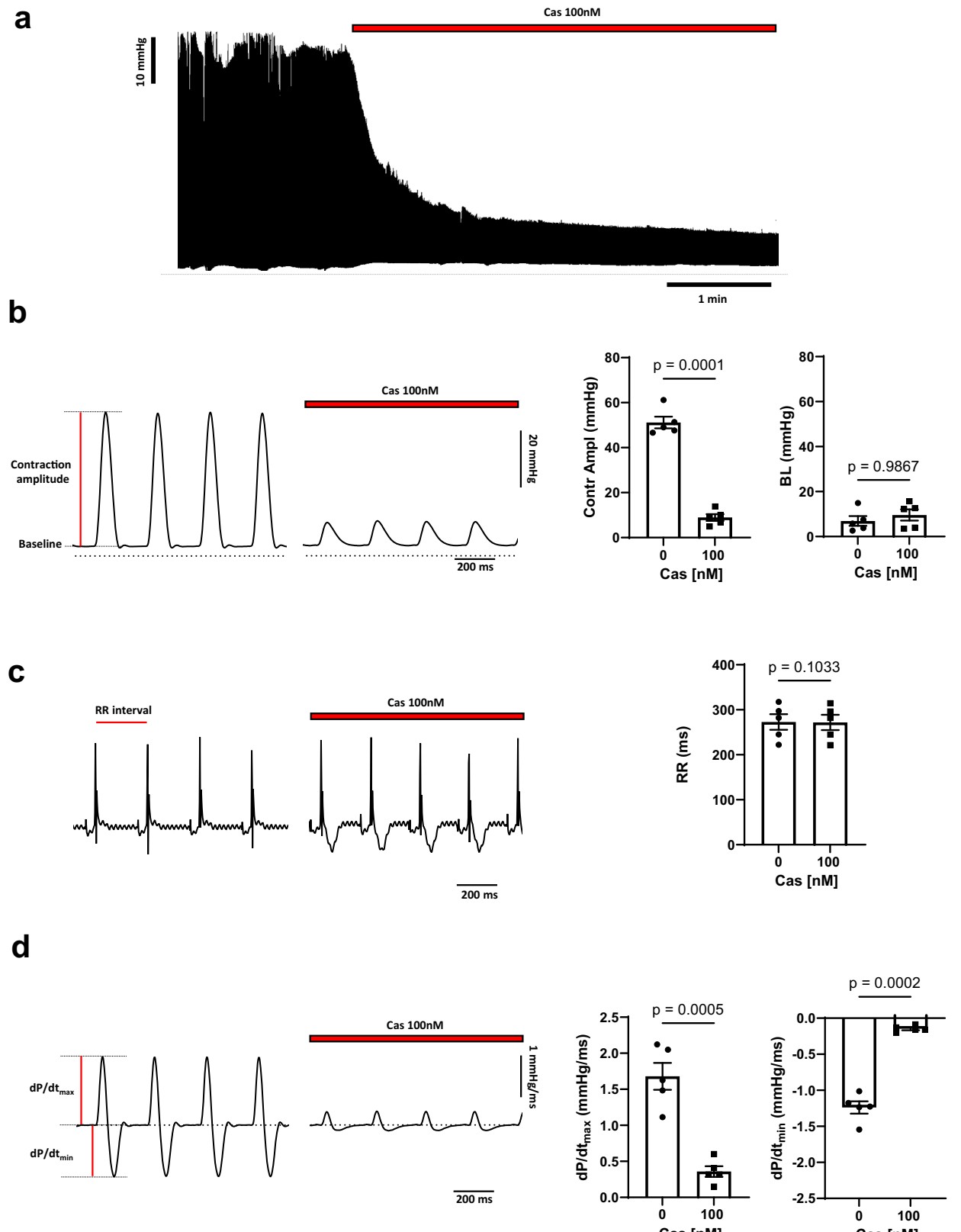

**Fig. 1 | Ventricular contraction and heart rate in Cas perfused heart. a** Time course of isolated perfused heart ($n = 5$) contraction amplitude recording before and after Cas 100 nM application. **b** Sample traces of pressure recordings (left) and histograms of left ventricular (LV) contraction amplitude and basal diastolic LV pressure before and after toxin perfusion ($n = 5$). **c** Representative ECG traces and averaged RR recorded from isolated perfused heart ($n = 5$) in ctrl condition and in presence of 100 nM Cas. **d** Sample traces of first derivative of LV pressure signal before and after toxin perfusion (left) and bar graphs showing inotropic (dP/dt$_{max}$) and lusitropic (dP/dt$_{min}$) negative effect of Cas 100 nM (right, $n = 5$). Statistics: paired two-sided Student $t$-test. Data are presented as mean values ± SEM.

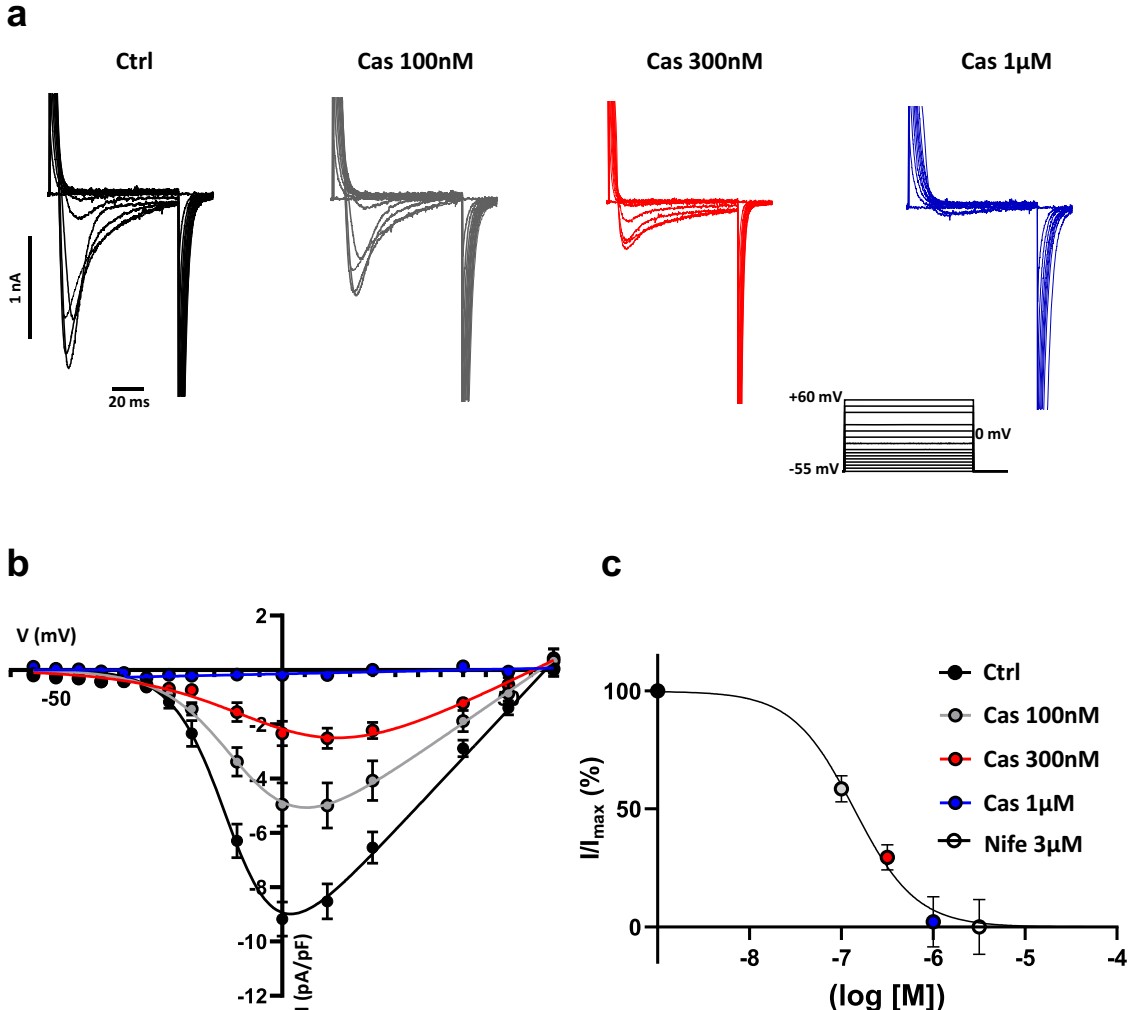

**Fig. 2 | Cas inhibition of L-type Ca²⁺ current (I$_{Cav}$1.2) in isolated ventricular myocytes. a** Representative Ca²⁺ current traces of the L-type Ca²⁺ current measured from a holding potential of −55 mV in Tyrode (black traces), or under Cas 100 nM (gray traces), Cas 300 nM (red traces), Cas 1 μM (blue traces) and Nife 3 μM (dotted black traces) perfusion in freshly isolated ventricular Wild-Type cardiomyocytes. Voltage clamp step protocol is reported in the inset of a. **b** Current-to-voltage (I–V) relationship of L-type Ca²⁺ current (Ca$_v$1.2) recorded from holding potential of −55 mV in ventricular cells before ($n = 22/N = 3$) and after 100 nM Cas

($n = 9/N = 3$), 300 nM Cas ($n = 7/N = 3$), 1 μM Cas ($n = 6/N = 3$) and 3 μM Nife ($n = 12/N = 3$) perfusion. Current densities are normalized to the maximal effect of nifedipine (a, blue traces). At peak: before vs 100 nM Cas, $p = 0.0071$; before vs 300 nM Cas, $p < 0.0001$; before vs 1 μM Cas, $p < 0.0001$. Statistics: one-way ANOVA test followed by Dunnet multiple comparisons test. Corresponding normalized dose-response of I$_{CaL}$ for peak current shown in (**c**). Data are presented as mean values ± SEM. $n$ = number of cells; $N$ = number of mice.

recorded I$_{CaL}$ from a holding potential of −55 mV at which I$_{CaT}$ is inactivated[4]. We thus recorded I$_{CaL}$ flowing through L-type Ca$_v$1.3 and Ca$_v$1.2 isoforms. In Wild-Type SAN myocytes, Cas (1 μM) reduced I$_{CaL}$ density at voltages ranging from −35 mV to +40 mV, which corresponded to the activation range of I$_{Cav1.2}$[8]. Residual I$_{CaL}$, namely Ca$_v$1.3-mediated current, was totally blocked by prototypical L-type DHP blocker nifedipine (3 μM)[20] (Fig. 4a, Supplementary Fig. 7). To further characterize the effect of the toxin on I$_{CaL}$, we used SAN myocytes isolated from Ca$_v$1.3$^{-/-}$ mice, in which the only residual I$_{CaL}$ is I$_{Cav1.2}$[8]. Cas (1 μM) totally blocked I$_{CaL}$ in isolated Ca$_v$1.3$^{-/-}$ SAN myocytes (Fig. 4b, Supplementary Fig. 8). We then measured, in Ca$_v$1.3$^{-/-}$ SAN myocytes, peak I$_{CaL}$ evoked by a voltage clamp step protocol at 0 mV from two different holding potentials, −60 mV and −40 mV respectively, in absence and after Cas 300 nM perfusion. We did not observe any statistically significant difference in the percentage of I$_{CaL}$ block ($68 ± 3\%$ vs $67 ± 4\%$ of total Ca$_v$1.2-mediated current for holding potential of −40 mV and −60 mV, respectively. Supplementary Fig. 9a) suggesting no voltage dependence of Cas block on I$_{Cav1.2}$. In addition, we evaluated the frequency dependence of Cas block by modifying the

frequency of the voltage clamp protocol (step to 0 mV from a holding potential of −60 mV). We did not observe any statistically significant difference in the degree of inhibition by 300 nM Cas in the three conditions tested ($62 ± 6\%$ at 0.2 Hz, $63 ± 6\%$ at 1 Hz and $58 ± 8\%$ at 5 Hz. Supplementary Fig. 9b). Taken together, these data show that Cas strongly discriminates between I$_{Cav1.2}$ and I$_{Cav1.3}$. The toxin selectively inhibits the Ca$_v$1.2 channels isoform, without showing dependency from holding potential or frequency of stimulation, indicating that inhibtion of I$_{Cav1.2}$ by Cas was not voltage- or use-dependent.

### Calciseptine reduces the action potential amplitude and spares automaticity in SAN pacemaker myocytes

To examine the effect of the toxin on SAN cell automaticity, we recorded spontaneous action potentials (APs) in SAN myocytes from Wild-Type mice using the perforated patch-clamp technique (Fig. 5a; Fig. 5g). Consistently, 300 nM and 1 μM Cas did not affect AP firing rate (Fig. 5b, h, Table 1), but reduced AP amplitude (Fig. 5c, i, Table 1). Concomitant perfusion of Cas 300 nM and nifedipine 3 μM (Fig. 5b, Table 1) or of Cas 1 μM and nifedipine 3 μM (Fig. 5h, Table 1)

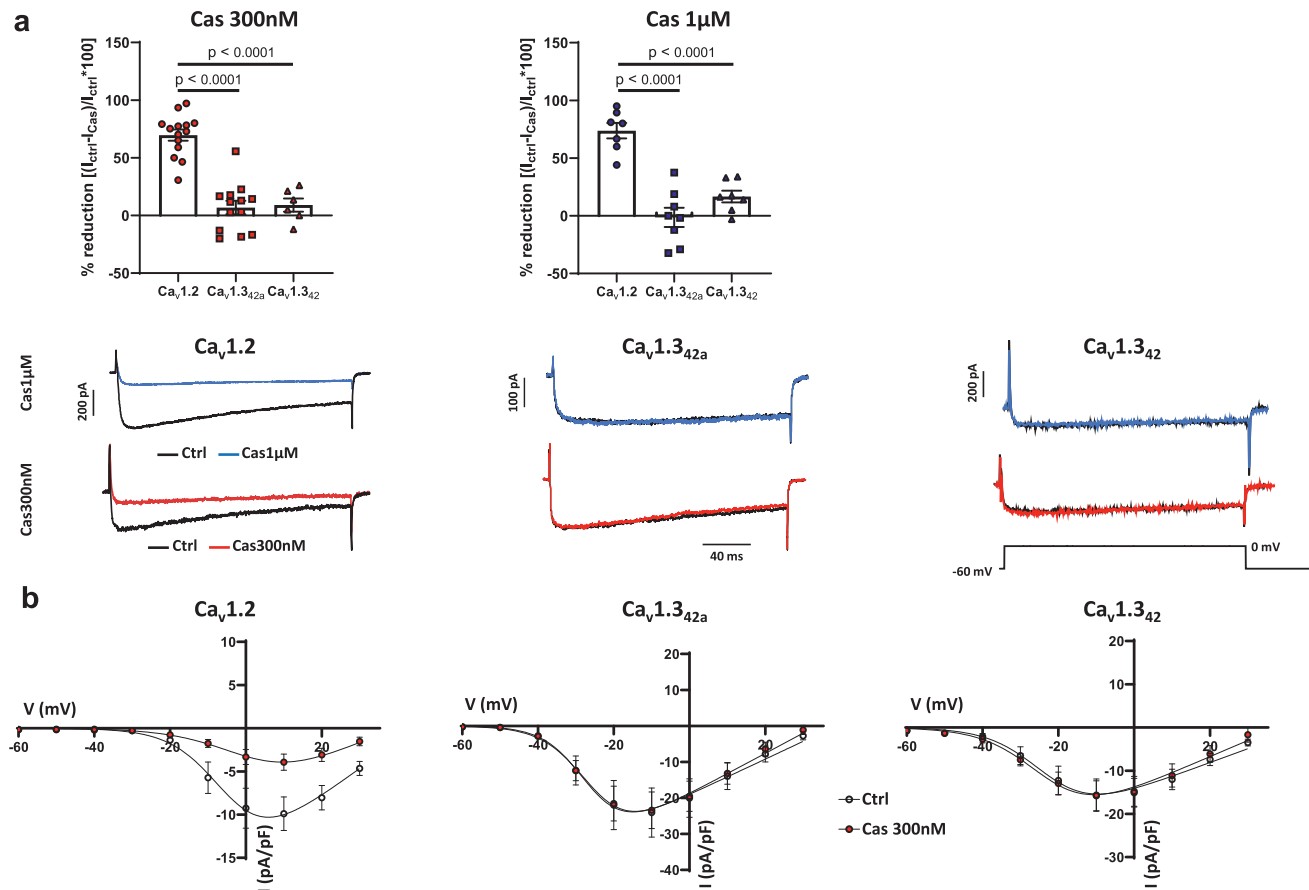

**Fig. 3 | Cas effect on L-type Ca²⁺ current density in HEK-293T cells expressing Ca_v1.2, Ca_v1.3_42a or Ca_v1.3_42 channels. a** Histograms represent the percentage of reduction of L-type Ca²⁺ peak current density after 300 nM and 1 μM Cas perfusion in HEK-293T cells transfected with Ca_v1.2 (circles, $n = 14/N = 4$ and $n = 7/N = 4$, respectively), Ca_v1.3_42a (squares, $n = 13/N = 4$ and $n = 8/N = 4$, respectively) or Ca_v1.3_42 (triangles, $n = 6/N = 3$ and $n = 7/N = 3$, respectively) channels. Peak current density was recorded starting from a holding potential of −60 mV using an activating voltage step at 0 mV. Examples of L-type Ca²⁺ current traces for each condition and for each clone tested are reported. Statistics: one-way ANOVA test followed by Tukey's multiple comparisons test. **b** Current-to-voltage (I−V) relationship of L-type Ca²⁺ current recorded from holding potential of −60 mV in HEK-293T Ca_v1.2 ($n = 11/N = 4$), Ca_v1.3_42a ($n = 9/N = 4$) and Ca_v1.3_42 ($n = 6/N = 3$) transfected cells before (empty circles) and after (red filled circles) 300 nM Cas application. At peak: for Ca_v1.2 before vs 300 nM Cas, $p = 0.0037$; for Ca_v3.1_42a before vs 300 nM Cas, $p = 0.8148$; for Ca_v1.3_42 before vs 300 nM Cas, $p = 0.5696$. Statistics: paired two-sided Student *t*-test. Data are presented as mean values ± SEM. $n =$ number of cells; $N =$ number of independent transfections.

induced a significantly decrease in the beating rate in SAN myocytes with respect either to control or to sole condition of Cas perfusion. Concomitant application of Cas and nifedipine 3 μM did not reduce AP amplitude further (Fig. 5c, i, Table 1). Slopes of linear (SLDD) and exponential (SEDD) part of diastolic depolarization did not change either following perfusion of Cas at 300 nM (Fig. 5d, e, Table 1) or at 1 μM (Fig. 5j, k, Table 1). Both SLDD and SEDD showed statistically significant decrease only when nifedipine 3 μM was added to the perfusion solutions (Fig. 5d, e and Fig. 5j, k, Table 1). Furthermore, we evaluated action potential duration in SAN myocytes exposed to Cas and Cas plus nifedipine 3 μM. While 300 nM Cas did not affect AP duration, we observed a reduction in action potential duration at 30%, 50% and 70% of repolarization only after concomitant application of Cas 300 nM and nifedipine 3 μM (Fig. 5f). By contrast, 1 μM Cas had statistically significant effect on AP duration at 30% and 50% of AP repolarization. Again, concomitant perfusion of Cas 1 μM and nifedipine 3 μM reduced AP duration at 30%, 50% and 70% of repolarization (Fig. 5l). We did not observe statistically significant differences in the maximum diastolic potential, the threshold potential and in action potential upstroke velocity of action potentials recorded in myocytes under control condition and after perfusion of Cas at either 300 nM or at 1 μM (Supplementary Fig. 10).

## Discussion

Our study shows that the black mamba venom toxin Cas selectively inhibits Ca_v1.2 channels underlying I_Cav1.2, without affecting I_Cav1.3 in the heart. Cas displays selectivity between the two cardiac of L-type Ca²⁺ channel subunits, enabling pharmacologic distinction between the functional roles of these L-type Ca²⁺ channel isoforms.

L-type Ca²⁺ channels are required for multiple functions in the heart, including regulation of early organogenesis, structural development, cardiac physiology and automaticity[21, 22]. A variety of arrhythmias including inherited QT syndrome[23] have been associated with alterations in L-type channels[4]. Consequently, Ca_v1 class channels members have been, are and will be prime targets for medical therapy[21, 24–26]. Pharmacological inhibition of Ca_v1.2 channels accounts for most of therapeutically relevant antihypertensive and anti-ischemic effects, predominantly by reducing peripheral vascular resistance and negative inotropy. Dihydropyridines (DHPs) are among the most used Ca_v1 channel antagonists[27] with a long history of success in the treatment of several cardiovascular pathologies[28, 29]. However, one potential limitation of DHPs in therapeutic perspective is their lack of selectivity for different isoforms of the Ca_v1 family. To date, over 100 DHPs were screened in an attempt to identify a selective antagonist able to distinguish among Ca_v1.2 and Ca_v1.3[30]. However, no candidate molecule issued from this screen study showed selectivity for

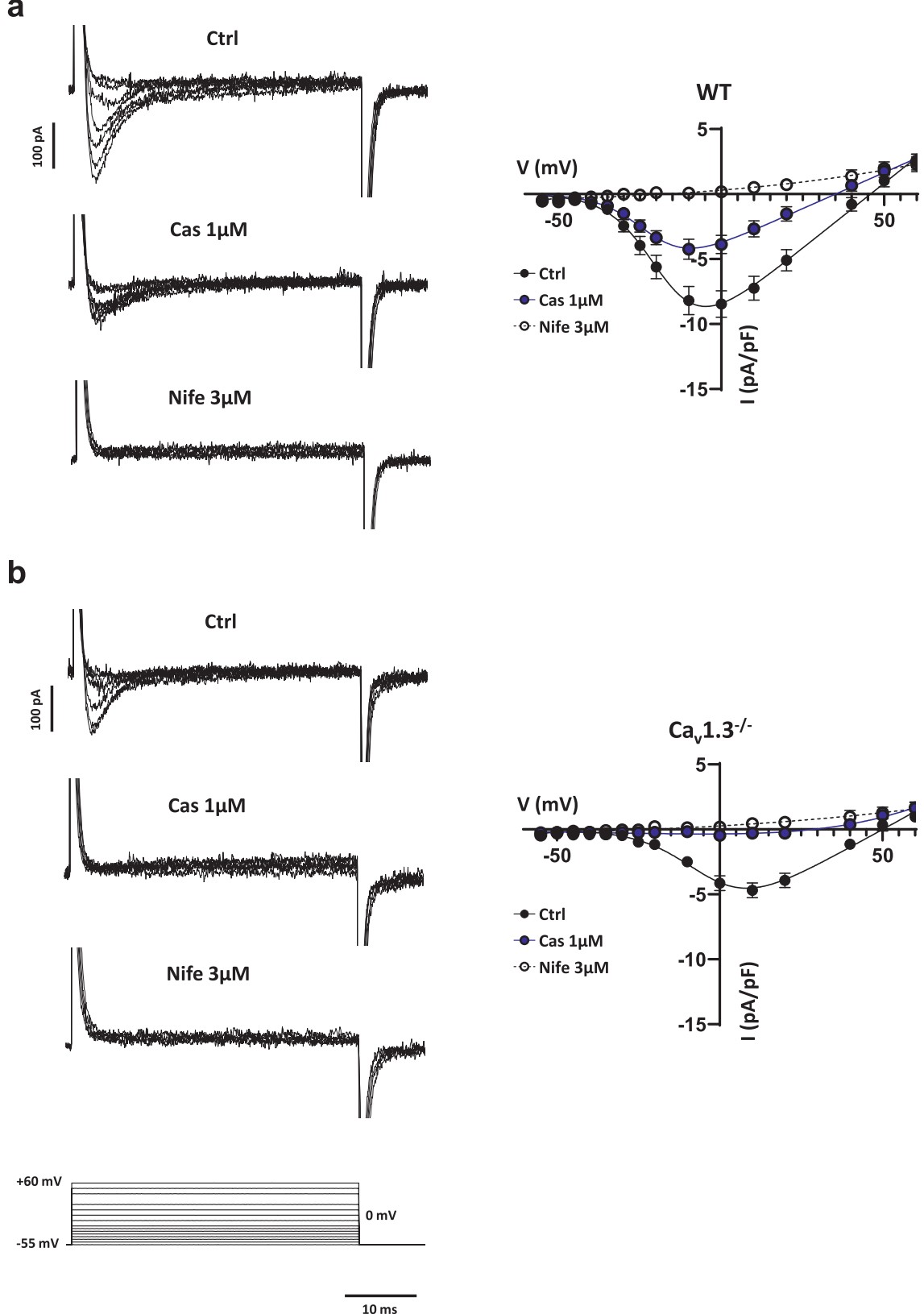

**Fig. 4 | 1 μM Cas effect on I$_{CaL}$ in isolated SAN cells from Wild-Type and Ca$_v$1.3$^{-/-}$ mutant mice. a** Current-to-voltage (I−V) relationship and sample traces of L-type Ca$^{2+}$ current recorded in isolated Wild-Type SAN cells ($n = 8/N = 2$) from a holding potential of −55 mV in control condition (black filled circles), after 1 μM Cas perfusion (blue circles) and after concomitant exposure to Cas 1 μM and Nife 3 μM (empty circles). At peak: before vs 1 μM Cas, $p = 0.0168$; 1 μM Cas vs Nife 3 μM,

$p = 0.0008$ Statistics: one-way ANOVA test followed by Tukey multiple comparisons test. **b** Same as in a but in isolated SAN myocytes form Ca$_v$1.3$^{-/-}$ ($n = 8/N = 2$). At peak: before vs 1 μM Cas, $p = 0.0018$; 1 μM Cas vs Nife 3 μM, $p = 0.1036$. Statistics: one-way ANOVA test followed by Tukey multiple comparisons test. Voltage-clamp step protocol used is shown at the bottom of the sample traces. Data are presented as mean values ± SEM. $n$ = number of cells; $N$ = number of mice.

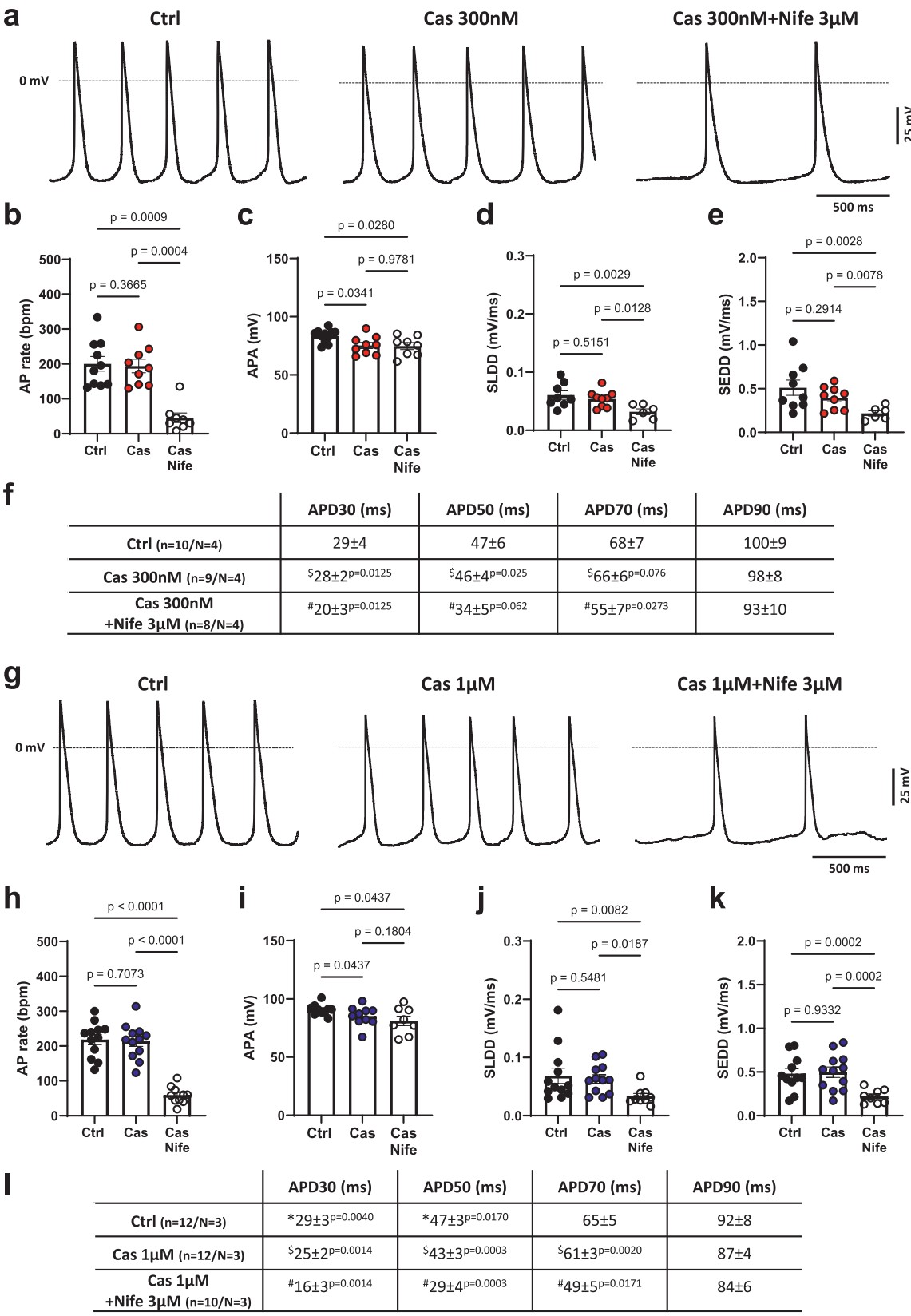

| | APD30 (ms) | APD50 (ms) | APD70 (ms) | APD90 (ms) |
|---|---|---|---|---|
| **Ctrl** (n=10/N=4) | 29±4 | 47±6 | 68±7 | 100±9 |
| **Cas 300nM** (n=9/N=4) | $28±2^{p=0.0125}$ | $46±4^{p=0.025}$ | $66±6^{p=0.076}$ | 98±8 |
| **Cas 300nM +Nife 3µM** (n=8/N=4) | #20±3$^{p=0.0125}$ | #34±5$^{p=0.062}$ | #55±7$^{p=0.0273}$ | 93±10 |

| | APD30 (ms) | APD50 (ms) | APD70 (ms) | APD90 (ms) |
|---|---|---|---|---|
| **Ctrl** (n=12/N=3) | *29±3$^{p=0.0040}$ | *47±3$^{p=0.0170}$ | 65±5 | 92±8 |
| **Cas 1µM** (n=12/N=3) | $25±2^{p=0.0014}$ | $43±3^{p=0.0003}$ | $61±3^{p=0.0020}$ | 87±4 |
| **Cas 1µM +Nife 3µM** (n=10/N=3) | #16±3$^{p=0.0014}$ | #29±4$^{p=0.0003}$ | #49±5$^{p=0.0171}$ | 84±6 |

one or the other isoform. This same study indicated highly similar DHP binding site at both $Ca_v1.2$ and $Ca_v1.3$ channels, suggesting that other classes of compounds needed to be identified to achieve selectivity[30]. Other potent synthetic blockers with completely different structure as compared to DHPs have been identified for their action on L-type $Ca^{2+}$ channels[31–33], but to the best of our knowledge, they have not been tested for their specificity for the different isoforms of $Ca_v1$ channels. Animal toxins have played an essential role in the physiological exploration of the bioelectric mechanisms of the heart and the brain. Over the last five decades, toxins have allowed investigation and characterization of the functional roles of several classes of ion channels; e.g. tetrodotoxin for the voltage-sensitive $Na^+$ channel[34], apamin

**Fig. 5 | Effect of Cas on automaticity in SAN pacemaker myocytes.**
**a** Representative action potential recordings in Wild-Type SAN cells in Tyrode, during Cas 300 nM and Cas 300 nM + Nife 3 μM perfusion. Dotted line indicates the 0 mV. Histograms represent the average rate (**b**), the amplitude (APA) (**c**), the slope of the linear (SLDD) (**d**) and of the exponential (SEDD) (**e**) part of the diastolic depolarization of spontaneous action potentials recorded in SAN cells in Tyrode solution (Ctrl, black filled circles, $n = 10/N = 4$), during 300 nM Cas perfusion (Cas, red filled circles, $n = 9/N = 4$) and in Cas 300 nM + Nife 3 μM solution (Cas+Nife, white circles, $n = 8/N = 4$). Statistics: one-way ANOVA test followed by Tukey's multiple comparisons test. **f** Action potential duration at 30%, 50%, 70% and 90% of the repolarization in control condition ($n = 10/N = 4$), during Cas 300 nM perfusion ($n = 9/N = 4$) and in Cas 300 nM + Nife 3 μM ($n = 8/N = 4$). *Ctrl vs Cas 300 nM, #Ctrl vs Cas 300 nM + Nife 3 μM, $Cas 300 nM vs Cas 300 nM + Nife 3 μM. Statistics: one-way ANOVA test followed by Tukey's multiple comparisons test. **g.** Representative

action potential recordings in Wild-Type SAN cells in Tyrode, during Cas 1 μM and Cas 1 μM + Nife 3 μM perfusion. Histograms represent the average rate (**h**), the amplitude (APA) (**i**), the slope of the linear (SLDD) (**j**) and of the exponential (SEDD) (**k**) part of the diastolic depolarization of spontaneous action potentials recorded in SAN cells in Tyrode SAN cells in Tyrode solution (Ctrl, black filled circles, $n = 12/N = 3$), during 1 μM CaS perfusion (Cas, blue filled circles, $n = 12/N = 3$) and in Cas 1 μM + Nife 3 μM solution (Cas+Nife, white circles, $n = 10/N = 3$). Statistics: one-way ANOVA test followed by Tukey's multiple comparisons test. (**l**). Action potential duration at 30%, 50%, 70% and 90% of the repolarization were calculated in $n = 12/N = 3$ SAN cells in control condition, in $n = 12/N = 3$ SAN cells during Cas 1 μM perfusion and in $n = 9/N = 3$ following Cas 1 μM + Nife 3 μM application. *Ctrl vs Cas 1 μM, #Ctrl vs Cas 1 μM + Nife 3 μM, $Cas 1 μM vs Cas 1 μM + Nife 3 μM. Statistics: one-way ANOVA test followed by Tukey's multiple comparisons test. Data are presented as mean values ± SEM. $n$ = number of cells; $N$ = number of mice.

**Table 1 | Action Potential parameters recorded in SAN pacemaker myocytes**

|  | Rate (bpm) | APA (mV) | SLDD (mV/ms) | SEDD (mV/ms) |
|---|---|---|---|---|
| Ctrl ($n = 10/N = 4$) | $200 \pm 21$ | *$83 \pm 2^{p=0.0341}$ | $0.06 \pm 0.01$ | $0.51 \pm 0.09$ |
| Cas 300 nM ($n = 9/N = 4$) | $\$194 \pm 19^{p=0.0004}$ | $75 \pm 3$ | $\$0.05 \pm 0.01^{p=0.0128}$ | $\$0.40 \pm 0.04^{p=0.0078}$ |
| Cas 300 nM +Nife 3 μM ($n = 8/N = 4$) | #$45 \pm 14^{p=0.0009}$ | #$75 \pm 3^{p=0.0280}$ | #$0.03 \pm 0.01^{p=0.0029}$ | #$0.21 \pm 0.03^{p=0.0028}$ |
| Ctrl ($n = 12/N = 3$) | $218 \pm 15$ | *$91 \pm 2^{p=0.0437}$ | $0.07 \pm 0.01$ | $0.48 \pm 0.06$ |
| Cas 1 μM ($n = 12/N = 3$) | $\$213 \pm 15^{p<0.0001}$ | $85 \pm 3$ | $\$0.06 \pm 0.01^{p=0.0187}$ | $\$0.50 \pm 0.06^{p=0.0002}$ |
| Cas 1 μM +Nife 3 μM ($n = 10/N = 3$) | #$59 \pm 88^{p<0.0001}$ | #$81 \pm 4^{p=0.0437}$ | #$0.03 \pm 0.01^{p=0.0082}$ | #$0.22 \pm 0.03^{p=0.0002}$ |

Rate, action potential rate; APA, action potential amplitude; SLDD, slope of linear part of diastolic depolarization; SEDD, slope of exponential part of diastolic depolarization. *Ctrl vs Cas 300 nM (or 1 μM), #Ctrl vs Cas 300 nM (or 1 μM) + Nife3μM, $Cas 300 nM (or 1 μM) vs Cas 300 nM (or 1 μM) + Nife3μM. Statistics: one-way ANOVA test followed by Tukey's multiple comparisons test.
Data are presented as mean values ± SEM. $n$ = number of cells; $N$ = number of mice.

for Ca$^{2+}$-sensitive K$^{+}$ channels[35], omega conotoxins for N-type Ca$^{2+}$ channels[36], psalmotoxin for ASIC channels[37]. Since the discovery of these leading toxins, tremendous information has accumulated on the molecular identification of L-type Ca$^{2+}$ channel isoforms in the brain, neuroendocrine and cardiovascular tissues. Thus, new pharmacological tools to dissect the roles played by different Ca$^{2+}$ channels that discriminate between isoforms belonging to the L-type Ca$_v$1 gene family are needed. This is of particular importance in the heart in which two L-type isoforms Ca$_v$1.2 and Ca$_v$1.3 could not be distinguished pharmacologically so far.

In this work, we investigated if Cas can discriminate between cardiac Ca$_v$1.2 and Ca$_v$1.3, and if potential selectivity could be used to study the respective physiological functions of Ca$_v$1.2 and Ca$_v$1.3 in determination of heart rate and contractility.

Our results show that Cas has selective negative inotropic effect on isolated perfused heart without affecting heart rate or inducing arrhythmia in isolated hearts. A previous study reported sinus bradycardia with unaltered contractile properties in isolated heart from Ca$_v$1.3$^{-/-}$ mice[38] suggesting that while Ca$_v$1.2 is the only L-type Ca$^{2+}$ channel isoform involved in heart contractility, Ca$_v$1.3 plays a major role in heart automaticity and conduction. In addition, the evidence that, contrary to what was observed in Ca$_v$1.3$^{-/-}$ mice, perfusion of Cas does not induce arhythmias, sinus bradycardia or heart block is in line with the hypothesis that Cas does not affect the I$_{Cav1.3}$. Consequently, these results indicate that pure negative inotropic effect of Cas is due to selective block of Ca$_v$1.2 vs Ca$_v$1.3 L-type Ca$^{2+}$ channels. We provide further evidence for selective Ca$_v$1.2 inhibition by Cas in SAN myocytes isolated from wild-type and Ca$_v$1.3$^{-/-}$ mice. Indeed, voltage clamp recordings in Wild-Type SAN myocytes showed that Cas account for around 45% reduction of total I$_{CaL}$. Considering that previous data reported that in SAN cells the I$_{Cav1.2}$ /I$_{Cav1.3}$ current density ratio is between 1:2[8] to 1:1[10], it is reasonable to postulate that the residual I$_{CaL}$ current recorded in SAN wild-type cells correspond to I$_{Cav1.3}$. This point

is supported by our recordings of I$_{CaL}$ in isolated Ca$_v$1.3$^{-/-}$ SAN myocytes before and after Cas perfusion. In control conditions I$_{CaL}$ peak current density in Ca$_v$1.3$^{-/-}$ SAN cells was quantitively similar to the amount of current blocked by Cas in Wild-Type SAN myocytes, further indicating that Cas acted selectively on I$_{Cav1.2}$. No residual I$_{CaL}$ was recorded in Ca$_v$1.3$^{-/-}$ SAN myocytes after Cas perfusion, confirming the role of Cas as pure Ca$_v$1.2 L-type Ca$^{2+}$ current blocker. Results obtained on recombinant L-type Ca$_v$1.3 currents mediated by short and long variants, T-type Ca$_v$3.1, N-type Ca$_v$2.2, P/Q-type Ca$_v$2.1 expressed in HEK-293T and on L-type Ca$_v$1.2 channels in isolated ventricular myocytes further strengthen this conclusion. Since Ca$_v$1.2 plays a major role in shaping the AP in cardiac myocytes[39] we studied the impact of Cas application on the different parameters in spontaneous SAN AP. Cas affects both AP amplitude and duration, without affecting the slope of diastolic depolarization phase which is known to be a Ca$_v$1.3-dependent feature[40], stressing again the efficiency and the selectivity of the toxin. Our results are consistent with previous studies on the functional role of Ca$_v$1.2 in the heart using global or conditional genetic ablation of Ca$_v$1.2 channels. Indeed, while these studies demonstrated obligatory role of this isoform in cardiac contraction[41,42] and regulation of blood pressure[43], they also reported lack of effect on heart rate.

The study by Teramoto and coll[44] on Cas effect on unitary Ba$^{2+}$ currents in the guinea-pig portal vein cells, suggested that the action site of Cas was outside the membrane. DHPs are prototypical organic blockers of L-type calcium channels widely used in therapeutical applications. A previous study reported that Cas at a rather high concentration could inhibit binding of DHPs in brain synaptosomes suggesting that the toxin is located in the proximity of DHP binding site[45].

To investigate if the binding site of Cas was overlapping with that of DHPs we used isolated ventricular myocytes from mice in which Ca$_v$1.2 channels have been rendered insensitive to DHPs by knock-in of a point mutation in the drug binding site (Ca$_v$1.2$^{DHP-/-}$)[46]. I$_{Cav1.2}$ recorded in Ca$_v$1.2$^{DHP-/-}$ myocytes showed similar sensitivity to Cas than that

those isolated from Wild-Type myocytes, suggesting that DHPs and CaS do not share the same binding site on $Ca_v1.2$ channels (Supplementary Fig. 11a). To futher explore the binding properties of Cas on $Ca_v1.2$, we also used an electrophysiological approach aimed to indicate if a toxin inhibiting a neuronal $Ca^{2+}$ channel was acting as a gating modifier[47]. We investigated the Cas-induced block of $Ca_v1.2$ expressed in HEK-293T cells at a moderate depolarizing potential (0 mV) and at a very positive potential (+150 mV) leading to maximal current activation. Our data show that Cas has almost no effect on outward-going $I_{Cav1.2}$, recorded at +150 mV−likely carried by $Cs^+$ ions−suggesting a gating modifier behavior. By contrast, using the same protocol, 3 μM Nifedipine partially inhibited the $I_{Cav1.2}$ elicited at +150 mV, again showing that Cas inhibits $Ca_v1.2$ channels with a different mechanism than DHPs (Supplementary Fig. 11b).

A very homologous snake venom peptide ($FS_2$) from black mamba was reported to have a similar action in muscle relaxation and hypotensive properties than Cas[48]. A common short peptide between two Prolines at position 42 and 47 (PTAMWP) was predicted as the binding site for Cas and $FS_2$ by Kini and co-workers[49]. These authors reported that this peptide called L-calchin acts like Cas and FS2 in reducing peak systolic pressure in isolated rat hearts and reducing calcium channel amplitude in rabbit cardiac myocytes but with lower potency ($IC_{50} \approx 5$ μM). Similar to Cas, block by L-calchin had no effect on heart rate in ex vivo rat isolated hearts. We thus synthetized L-calchin and conducted preliminary experiments on dissociated mouse ventricular myocytes (Supplementary Fig. 12). Our results showed ≈30% reduction of $I_{Cav1.2}$, in line with these previous data. Taken together, our data and comparison with previous studies on this class of peptides indicate that Cas and Cas-derived peptides are a promising starting point to develop selective inhibitors of $Ca_v1$ channel subtypes.

L-type $Ca^{2+}$ channels are widely expressed in the brain and, in many neuronal types, $Ca_v1.2$ and $Ca_v1.3$ are co-expressed (see ref. 50 for review)[50]. Both distinct and complementary functions for $Ca_v1.2$ and $Ca_v1.3$ have been proposed in the central nervous system. For example, $Ca_v1.2$ is involved in hippocampal firing[51], spatial memory and LTP[52], as well as in the myelinisation process[53]. It is very likely that Cas, with its high specificity for the $Ca_v1.2$ channel, and its high flexibility of use due to the fact that it can be stereotoxally injected in specific sites in the brain will constitute a very useful tool for future explorations of the role of $Ca_v1.2$ in the nervous system. Beside playing a major role in cardiac pacemaker activity, $Ca_v1.3$ appear to be an important player in coupling spontaneous neuronal firing with mitochondrial metabolism[54]. It has been proposed that $Ca_v1.3$ channels are involved in firing-dependent oxidative stress of dopaminergic neurons of *substantia nigra*, suggesting that these channels could constitute a potential target for Parkinson disease[55]. This has sparkled studies in the search for selective $Ca_v1.3$ antagonists (for review see[1, 56]). Recently, a molecule named compound 8 was described as selective for $Ca_v1.3$[57] with affinity ~24 μM. However, the selectivity of compound 8 is controversial[58, 59]. In particular, Huang et al. 2014[58] reported that selectivity of compound 8 for $Ca_v1.3$ versus $Ca_v1.2$ is modest and dependent on the nature of the associated β subunit and possibly dependent from the $Ca_v1.3$ splice variants. It will thus be interesting to study selectivity of Cas in neurons coexpressing $Ca_v1.2$ and $Ca_v1.3$.

## Methods
### Care and use of animals
The study is in accordance with the Guide for the Care and Use of Laboratory Animals published by the US National Institute of Health (NIH Publication No. 85-23, revised 1996) and European directives (2010/63/EU). Experimental procedures were approved by the Ethical Panel of the University of Montpellier and the French Ministry of Agriculture (protocol no: 2017010310594939). Animals were housed in the IGF animal facility with free access to food and water and were exposed to 12 h light/dark cycles in a thermostatically controlled room (21–22 °C) with 40−60% humidity.

### Whole heart contractility
Hearts were isolated from adult male mice anaesthetized with a first intraperitoneal injection of a mixture comprising ketamine (100 mg/kg) and xylazine (10 mg/kg), and a second one of sodium pentobarbital (45 mg/kg; Ceva Santé Animale, France). An intraperitoneal injection of heparin (7500IU/kg; Sanofi Aventis, France) was administered in order to avoid thrombus formation. A thoracotomy was performed and beating hearts were quickly removed and excised into ice-cold modified Krebs-Henseleit buffer containing (mM): NaCl 116; KCl 5; $MgSO_4$ 1.1; $NaH_2PO_4$ 0.35; $NaHCO_3$ 27; $CaCl_2$ 1.8 and glucose 10. Buffer was gassed with 95% $O_2$ and 5% $CO_2$, giving a pH of 7.4 at 37 °C. Before use, the buffer was initially passed through a 0.2 μm filter (Sarstedt, USA). The aorta was cannulated under magnification and hearts were quickly mounted and perfused on the Langendorff perfusion system (Isolated heart system, EMKA Technologies, France). The coronary circulation was perfused at a constant pressure (70 mmHg) over a 1 to 4 ml/min flow range with modified Krebs-Henseleit buffer. During the stabilization phase, the heart was continuously perfused with modified Krebs-Henseleit and the electrocardiogram (ECG) was recorded through Ag/AgCl electrodes positioned on the right atrium and near the apex. Heart temperature was of 37 °C as continuously assessed by a probe positioned on the base of the heart and recorded by using a TH-5 monitoring thermometer (Phymep, USA). LV pressure was continuously measured by a home-made balloon inserted into the LV cavity by the left atrium and connected to the EMKA amplifier. After 20 min of stabilization (baseline), 100 nM Cas was perfused into the heart for 7 min. Then, the modified Krebs-Henseleit buffer was perfused for 1 h allowing to wash the effects of the Cas toxin. During the whole ex vivo protocol, the left ventricular pressure signals were continuously digitized at 0.1 Hz and recorded using an IOX data acquisition system (EMKA Technologies, France). Data were analyzed using ecgAUTOv3.3.3.12 (EMKA technologies) and MATLAB-R2021b (The MathWorks) custom-made script

### Cell culture and transfection protocols
HEK-293T (RRID:CVCL_2737) were obtained from the European Collection of Authenticated Cell Cultures (ECACC 96121229). The identity of HEK-293T has been confirmed by STR profiling and the cells have been eradicated from mycoplasma at ECACC. We routinely tested the cells for the absence of the mycoplasma contamination. Cells were cultivated at 37 °C in DMEM supplemented with GlutaMax, 10% fetal bovine serum and 1% penicillin/streptomycin (Invitrogen, Fisher Scientific, France). Transfections were performed using jet-PEI (Ozyme, France) with a DNA mix containing either the plasmids that code for $Ca_v1.2$-GFP, β2a and α2δ (0.8 μg of each) or either the plasmids that code for $Ca_v1.3$ (with and without exon 42a), or $Ca_v2.1$, or either $Ca_v2.2$ combined with β2a and α2δ (0.8 μg of each supplemented with 0.025 μg of a plasmid coding for GFP). For $Ca_v3.1$ expression β2a and α2δ subunits were omitted. Plasmids coding for β2a, α2δ and $Ca_v1.3$ were created as described previously[58]. Plasmid coding for $Ca_v1.2$-GFP was obtained from Dr. Catalucci[60]. After transfection cells were placed at 30 °C for 2−3 days and dissociated with Versene (Invitrogen, Fisher Scientific, France) and plated at a density of ~$35 \times 10^3$ cells per 35 mm Petri dish for electrophysiological recordings, which were performed the following day.

### Xenopus oocyte preparation, DNA injection and electrophysiology
*Xenopus laevis* specimen were purchased from Tefor Paris Saclay. Animal handling and experiments fully conformed to French regulations and were approved by local governmental veterinary services (authorization number E06-152-5 delivered by the Ministère de

l'Agriculture, Direction des Services Vétérinaires). Animals were anesthetized by exposure for 20 min to a 0.1% solution of 3-aminobenzoic acid ethyl ester (MS-222) (Sigma) buffered at pH 7.4. Oocytes were surgically removed and dissected away in a saline solution (ND96) containing (in mM) 96 NaCl, 2 KCl, 1.8 CaCl2, 2 MgCl2, and 5 HEPES at pH 7.4 with NaOH. Stage V and VI oocytes were treated for 3 h with collagenase (1 mg/ml, type Ia, Sigma) in the presence of trypsin inhibitor (Sigma) in ND96 to discard follicular cells. pSI-HERG plasmid solutions were injected in nucleus (100 ng/µl, 23 nl/oocyte) using a pressure microinjector. The oocytes were kept at 19 °C in the ND96 saline solution supplemented with penicillin (10 mg/L) and streptomycin (20 mg/L). Oocytes were studied within 2 days following injection of plasmid. In a 0.3-ml perfusion chamber, a single oocyte was impaled with two standard glass microelectrodes (1–2.5 MΩ resistance) filled with 3 M KCl and maintained under voltage- clamp using a Dagan TEV 200 amplifier. Stimulation of the preparation, data acquisition and analysis were performed using pClamp software (Axon Instruments). All experiments were performed at controlled room temperature (19-20 °C) in ND96 solutions. A perfusion system allowed local drugs external application. HERG currents were evoked every 10 s by 2.5 s depolarizing voltage steps ranging from -80 to +80 mV in 20 mV increments and then by a repolarizing step to -40 mV for 1.5 s. BeKm1 peptide was purchased from Smartox (Saint Martin d'Hères, France).

### Isolation of adult ventricular cardiomyocytes

Excised hearts from C57Bl/6 J Wild-Type and mice carrying DHP insensitive $Ca_v1.2$ channels ($Ca_v1.2^{DHP-/-}$) mice were quickly mounted on a Langendorff apparatus. Heart was perfused with a $Ca^{2+}$-free Tyrode solution for 4 min followed by an enzymatic solution containing LiberaseTM (0.2 mg/ml) and Trypsin (0.14 mg/ml) for 3.5 min. Hearts were removed from the apparatus and stored in a "Stop solution" (10 mM BDM, 5.5 mM glucose, 12.5 µM CaCl2 and 5% Fetal Calf Serum) to block enzymatic digestion. Atria were quickly removed and the ventricles were chopped into small pieces (~1 mm³) using forceps and dissociated by gentle pipetting using a flame-forged Pasteur pipette. Cell suspension was then filtered using a medical gauze and myocytes gravity settle for 20 min. To restore $Ca^{2+}$ concentration to physiological level, a $Ca^{2+}$ reintroduction buffer containing (mM): NaCl, 140; KCl, 5.4; MgCl2, 1; CaCl2, 1.8; Hepes,5 and glucose, 5.5 (pH adjusted to 7.4 with NaOH) was added progressively. Then cardiomyocytes were plated on dishes for patch clamp recording.

### Isolation of SAN cells

Hearts from Wild-Type and $Ca_v1.3^{-/-}$ mice were removed under general anesthesia consisting of 0.01 mg/g xylazine (2% Rompun; Bayer AG) and 0.1 mg/g ketamine (Imalgène;Merial) and immersed in a pre-warmed (36 °C) Tyrode solution containing (mM): 140 NaCl, 5.4 KCl, 1 MgCl2, 1.8 CaCl2, 5.5 D-glucose, and 5 Hepes (adjusted to pH 7.4 with NaOH). SAN region was identified using the superior and inferior vena cava, the crista terminalis, and the interatrial septum as landmarks. SAN tissue was excised and transferred into a low-$Ca^{2+}$ solution containing (mM): 140 NaCl, 5.4 KCl, 0.5 MgCl2, 0.2 CaCl2, 1.2 KH2PO4, 50 taurine, 5.5 D-glucose, 1 mg/mL BSA, and 5 Hepes–NaOH (adjusted to pH 6.9 with NaOH) for 5 min. Then, enzymatic digestion was carried out for 20-25 min at 36 °C in the low-$Ca^{2+}$ solution containing LiberaseTH Research Grade (0.15 mg/mL; Roche, Mannheim, Germany) and elastase (0.5 mg/mL; Worthington, Lakewood, NJ, USA). Digestion was stopped by washing the SAN in a "Kraftbrühe" (KB) medium containing (mM) 100 K-glutamate, 10 K-aspartate, 25 KCl, 10 KH2PO4, 2 MgSO4, 20 taurine, 5 creatine, 0.5 EGTA, 20 D-glucose, 5 Hepes, and 1 mg/mL BSA (adjusted to pH 7.2 with KOH). Single SAN cells were then dissociated from the SAN tissue by manual agitation using a flame-forged Pasteur's pipette. To recover the automaticity of the SAN cells, $Ca^{2+}$ was gradually reintroduced in the cell's storage solution to a final concentration of 1.8 mM. SAN cells were then left to rest for 1 h before recordings.

### Electrophysiological recordings

Macroscopic currents in HEK-293T cells were recorded at room temperature using an Axopatch 200B or Multiclamp 700B patchclamp amplifier (Molecular Devices, Sunnyvale CA). Borosilicate glass pipettes had a resistance of 1.5–2.5 MΩ when filled with an internal solution containing (in mM): 140 CsCl, 10 EGTA, 10 HEPES, 3 Mg-ATP, 0.6 GTPNa, and 3 CaCl2 (pH adjusted to 7.25 with KOH, ~315 mOsm, ~100 nM free $Ca^{2+}$ using the Max-Chelator software, http://maxchelator.stanford.edu/). The extracellular solution contained (in mM): 135 NaCl, 20 TEACl, 5 BaCl2, 1 MgCl2, and 10 HEPES (pH adjusted to 7.25 with KOH, ~330 mOsm).

For patch-clamp recordings of SAN myocytes, cells were harvested in custom-made chambers with glass bottoms for cell attachment. Cells were then super-fused with Tyrode's solution warmed at 36 °C before recording. Action potentials and ionic currents were recorded using an Axon multi-clamp patch-clamp 700B amplifier. Voltage-gated L-type $Ca^{2+}$ currents ($I_{CaL}$) were recorded using standard whole cell patch-clamp configuration. For recording $I_{CaL}$, we used an extracellular recording solution contained (mM): 135 tetraethylammonium chloride (TEA-Cl), 10 4-aminopyridine (4-AP), 1 MgCl2, 0.03 tetrodotoxin (TTX), 1 g/L Glucose, 2 CaCl2, 10 Hepes, (adjusted to pH = 7.2 with TEAOH). Recording electrodes, pulled from borosilicate glass using a DMZ-Universal Electrode Puller (Zeitz Instruments), had a resistance of 3MΩ and were filled with a solution containing (in mM): 125 CsOH, 20 TEA-Cl, 1.2 CaCl2, 5 Mg-ATP, 0.1 Li2-GTP, 5 EGTA, and 10 HEPES (pH adjusted to 7.2 with CsOH). Pacemaker activity was recorded by the perforated patch-clamp technique using escin (40 µg/ml). For recording cell automaticity glass pipettes were filled with the intracellular solution contained (mM): 80 K-Aspartate, 50 KCl, 1 MgCl2, 5 HEPES, 2 CaCl2, 5 EGTA, 3 ATP-Na+ salt; pH was adjusted to 7.2 with KOH. Escin was dissolved in sterile $H_2O$ and added to the intracellular solution right before each experiment.

We used an extracellular recording solution (Tyrode's based) contained (mM): 140 NaCl, 5.4 KCl, 1 MgCl2, 1.8 CaCl2, 5 HEPES, 5.5 D-Glucose; pH was set to 7.4 with NaOH. AP parameters have been calculated as reported in Baudot et al.[10]. Data were analyzed using pCLAMP9 (Molecular Devices) and GraphPad Prism (GraphPad) software.

### Toxin Calciseptine

In all experiments of cellular electrophysiology, we employed synthetized analog of Cas, commercially available at Latoxan (ID: L8109). The native peptide was purified from the venom of Dendroaspis polylepis purchased from Latoxan, as previoulsy described[16]. The pure peptide was obtained after gel filtration, cation exchange and C18 reversed phase chromatography. Its measured molecular mass (7036.26 Da average) was in accordance with the mass calculated from sequence data (7036.22 Da).

### Statistical analysis

Statistical analysis was performed using Prism 10.0 (GraphPad Software). Data are represented as mean ± the standard error of the mean. $p < 0.05$ was considered statistically significant. Normality test was performed on each data set and appropriate statistical tests have been applied. Statistical tests used in each experiment are specified throughout the figure legends.

### Reporting summary

Further information on research design is available in the Nature Portfolio Reporting Summary linked to this article.

## Data availability
All data that support the findings of this study are included in the manuscript. Original files of electrophysiological recordings and raw data are available upon request to the corresponding authors. A source data file is included with this manuscript. Source data are provided with this paper.

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

## Acknowledgements

We thank Jörg Striessnig for sharing the $Ca_v1.3^{-/-}$ and $Cav1.2^{DHP-/-}$ mouse lines. This work has been supported by the *Fondation Leducq* (TNE FANTASY 19CV03 to M.E.M). The IGF group is a member of the Laboratory of Excellence Ion Channel Science and Therapeutics supported by a grant from ANR (ANR-11-LABX-0015). The IGF group thanks the Réseau d'Animaleries de Montpellier (RAM) of Biocampus facility for the management of mouse lines. We thank all the personnel of the PCEA mouse breeding facility and of iExplore facility in Montpellier.

## Author contributions

P.M., M.E.M. and J.N. designed the study and wrote the manuscript. P.M., J.N., J.C., C.B, E.T., L.G., A.M., S.D. and I.B. performed experiments. P.M, J.N, J.C. analyzed the results. S.D., M.L., T.W.S. and S.B.-L. contributed to study design and conceptualization. All authors contributed to data interpretation and manuscript review and editing.

## Competing interests

The authors declare no competing interests
