## [Peer Review File · Nature Communications]

Selective blockade of Cav1.2 ($\alpha 1C$) versus Cav1.3 ($\alpha 1D$) L-type calcium channels by the black mamba toxin calciseptineReviewers' Comments:

Reviewer #1:

Remarks to the Author:

CaV1.2 and CaV1.3 channels play crucial roles in mediating Ca²⁺ influx in the heart and the brain. Although highly homologous, the channels have biophysical properties that are thought to enable distinct physiological functions. While pharmacological blockers of CaV1.2/1.3 have long-existed, there are no available approaches to distinguish between the two in the physiological setting. In this study, the authors probed the selectivity of a snake venom toxin, calciseptine (cas), in blocking CaV1.2 versus CaV1.3. Previous studies have shown that cascan eliminate cardiac contractions. Here the authors confirmed this finding and further demonstrated that cas does not alter automaticity. Based on this, the authors show that cas is high affinity inhibitor of CaV1.2 but not Cav1.3. The authors further illustrates the utility of this tool by testing how selective block of Cav1.2 affect SAN pacemaker function. Overall the study is very well-conducted and the findings are of broad relevance to the ion channel and calcium signaling community. Calciseptine will likely be used for a wide range of neuroscience and cardiovascular applications.

I have only minor concerns as noted below.

1. The authors note the possibility of engineering cas further for reversing selectivity. However, the mechanism of cas function is conspicuously absent. Are there any hints on how this toxin accomplishes selectivity? could it interact with the pore or alter voltage sensing domains or other domains?

2. The electrophysiology traces in Fig. 3A (both cas 1 μ m and 300 nm) looks like it is cutoff during the activating pulse. This should be replaced, the exemplar current should at minimum show the full trace even if the tail / gating current is cutoff.

3. Although the study nicely very illustrates the selectivity for CaV1.2 versus CaV1.3, its effect on other Cav channels is, however, not clear. Other peptide toxins (e.g. Protoxin II) often affect multiple distantly related channel types despite being "selective." It may be helpful to more systematically characterize effect on other Cav channels at least at one concentration. Indeed, there is some evidence that cas can block multiple ca channel types (Teramoto et al Pflugers Archive 432:462 (1996)).

Reviewer #2:

Remarks to the Author:

Mesirca et. al. present a study demonstrating a selective block of the CaV1.2 vs. CaV1.3 by the toxin calciseptine (Cas). Such selectivity has been long sought in the field and enables previously unattainable resolution of each L-type channel current in isolation. The study demonstrates a clear difference in block on both recombinant channels, and in the context of both ventricular myocytes and SA nodal cells. The toxin is able to reduce the action potential amplitude of SA nodal cells without effecting the firing rate, demonstrating an ability to differentially alter functional in a manner expected for a CaV1.2 selective blocker. The authors further show that, in contrast to many other calcium channel blockers, the toxin does not display frequency dependent block. Overall, the results are robust and significant. The data is clear and strongly supports the conclusions. I have only two questions for the authors to consider.

1. The authors demonstrate a clear difference between toxin block of CaV1.2 vs. CaV1.3, however given the propensity for other calcium channel blockers to effect hERG, it would be useful to know if this toxin has any effect on this channel type?

2. The splice variant of CaV1.3 used in the recombinant experiments is the exon 42a version, corresponding to a shortened form of the channel. It might be useful to confirm that the toxin also lacks the ability to block the long CaV1.3 isoform.

REVIEWER COMMENTS

Reviewer #1 (Remarks to the Author):

CaV1.2 and CaV1.3 channels play crucial roles in mediating Ca²⁺ influx in the heart and the brain. Although highly homologous, the channels have biophysical properties that are thought to enable distinct physiological functions. While pharmacological blockers of CaV1.2/1.3 have long-existed, there are no available approaches to distinguish between the two in the physiological setting. In this study, the authors probed the selectivity of a snake venom toxin, calciseptine (cas), in blocking CaV1.2 versus CaV1.3. Previous studies have shown that cas can eliminate cardiac contractions. Here the authors confirmed this finding and further demonstrated that cas does not alter automaticity. Based on this, the authors show that cas is high affinity inhibitor of CaV1.2 but not Cav1.3. The authors further illustrates the utility of this tool by testing how selective block of Cav1.2 affect SAN pacemaker function. Overall the study is very well-conducted and the findings are of broad relevance to the ion channel and calcium signaling community. Calciseptine will likely be used for a wide range of neuroscience and cardiovascular applications.

I have only minor concerns as noted below.

We thank the reviewer for the positive evaluation of our manuscript and for constructive comments and suggestions to improve our research work. See below our point-by-point responses.

Q1. The authors note the possibility of engineering cas further for reversing selectivity. However, the mechanism of cas function is conspicuously absent.

R1. The reviewer is right. We have deleted the following sentence 'Finally, it is possible that Cas derived peptides could be designed to reverse selectivity and create prototypes of Cav1.3 blockers.' which suggests premature possibility.

Q2. Are there any hints on how this toxin accomplishes selectivity? could it interact with the pore or alter voltage sensing domains or other domains?

R2. We thank the reviewer for raising this point. A dedicated study would be needed to answer to this interesting question properly. We think that at this point, even if important, is not among the aims of the present study and it would require a fully dedicated time for further research. However, in order to pave the way and suggest a valid hypothesis for future studies investigating the binding properties of Cas on Ca_v1.2 we conducted two series of experiments. At first, we performed a pilot study using isolated ventricular myocytes from mice in which Ca_v1.2 channels have been rendered insensitive to DHPs by knock-in of a point mutation in the drug binding site (Ca_v1.2^{DHP^{-/-}}). *I*_{Ca_v1.2 recorded in Ca_v1.2^{DHP^{-/-}} myocytes showed similar sensitivity to Cas than that of wild-type myocytes, suggesting that DHPs and Cas do not share the same binding site to Ca_v1.2 channels (Suppl. Fig. 11A). To further investigate the binding properties of Cas we focused on Cas-induced block of Ca_v1.2 (expressed in HEK-293T cells) in response to a moderate depolarizing potential (0 mV) and a very positive potential (+150 mV) that led to maximal current activation and switched the current to outward direction. This electrophysiological protocol was reported in the literature as a way to evaluate if Cas affects *I*_{Ca_v1.2 by altering the voltage-dependent gating of channel (McDonough et al, Journal of General Physiology 2002; 119:313–328; Bourinet and Zamponi, Neuropharmacology 2017 Dec;127:109-115.). In this experiment, the Ca_v1.2 current was alternatively elicited at 0 mV and at +150 mV (supplementary fig 11B). We found that 300nM Cas has no effect on the Ca_v1.2 current}}

(outward) at maximal activation potential (+150 mV) while the $Ca_v1.2$ current elicited at 0 mV (inward) is almost fully inhibited. This result suggests that Cas alters the channel gating properties of $Ca_v1.2$ and this would be compatible with a gating modifier mechanism.

This point deserves further specific investigations and additional structural studies are required to elucidate the interaction and binding properties between Cas and $Ca_v1.2$ channel. In contrast, $3\mu\text{M}$ Nifedipine inhibited outward-going $I_{Ca_v1.2}$ recorded at +150 mV, further supporting the hypothesis that Cas and DHPs interact with $Ca_v1.2$ channel with distinct mechanisms (Suppl. Fig. 11B). All these data are now included in the revised version of the paper. We are aware that these results do not entirely bridge the gap of knowledge about Cas binding site and selectivity, yet they constitute promising starting point for further investigations.

2. The electrophysiology traces in Fig. 3A (both cas 1 μm and 300 nm) looks like it is cutoff during the activating pulse. This should be replaced, the exemplar current should at minimum show the full trace even if the tail / gating current is cutoff.

Examples traces in Figure 3A have been replaced

3. Although the study nicely very illustrates the selectivity for $Ca_v1.2$ versus $Ca_v1.3$, its effect on other Cav channels is, however, not clear. Other peptide toxins (e.g. Protoxin II) often affect multiple distantly related channel types despite being "selective." It may be helpful to more systematically characterize effect on other Cav channels at least at one concentration. Indeed, there is some evidence that cas can block multiple ca channel types (Teramoto et al Pflugers Archive 432:462 (1996)).

We have performed new experiments on other Ca^{2+} channels isoforms, namely $Ca_v2.2$, $Ca_v2.1$ and $Ca_v3.1$ as well as on the long form of $Ca_v1.3$ (see point 2, reviewer2). We failed to detect inhibitory effect of Cas, both at 300nM and $1\mu\text{M}$, on N-type $Ca_v2.2$, on P/Q-Type $Ca_v2.1$ and on T-Type $Ca_v3.1$ mediated currents. We have added these results in the revised version of the manuscript (Suppl. Fig. 5).

In the paper of Teramoto (Teramoto et al, Pflugers Arch 1996;432(3):462-70), authors identified (in the cell-attached condition with 90 mM Ba^{2+} solution in the pipette and high K^+ solution in the bath), two different DHP-sensitive Ca^{2+} channel conductances in smooth muscle cells of the guinea-pig portal vein: 25-pS and 12-pS channels. Both conductances were blocked by Cas. Since $Ca_v1.2$ is the major L-type channel expressed in portal vein ($Ca_v1.3$ is not expressed in vascular tissue), these two conductances probably rely on this channel. Indeed, the unitary conductance depends on ion channel splice variants, on auxiliary subunit composition as well as specific channel regulations that occur in native tissues (Ghosh et al, Advances in Pharmacology 2017; 78: 49-87; Schjött et al, J Biol Chem 2003;278(36):33936-42; Neely et al, Am J Physiol 1995; 268(3Pt1):C732-40). Therefore, we think that the results of Teramoto et al. are fully in accord with our study. This interesting paper is now cited in the revised version of the manuscript

Reviewer #2 (Remarks to the Author):

Mesirca et. al. present a study demonstrating a selective block of the $Ca_v1.2$ vs. $Ca_v1.3$ by the toxin calciseptine (Cas). Such selectivity has been long sought in the field and enables previously unattainable resolution of each L-type channel current in isolation. The study demonstrates a clear difference in block on both recombinant channels, and in the context of both ventricular myocytes and SA nodal cells. The toxin is able to reduce the action potential amplitude of SA nodal cells without effecting the firing rate, demonstrating an ability to differentially alter functional in a manner expected for a $Ca_v1.2$ selective blocker. The authors further show that, in contrast to many

other calcium channel blockers, the toxin does not display frequency dependent block. Overall, the results are robust and significant. The data is clear and strongly supports the conclusions. I have only two questions for the authors to consider.

Q1. The authors demonstrate a clear difference between toxin block of CaV1.2 vs. CaV1.3, however given the propensity for other calcium channel blockers to effect hERG, it would be useful to know if this toxin has any effect on this channel type?

R1. We are grateful to the reviewer for giving us the opportunity to clarify this point. We performed experiments in *Xenopus* oocytes expressing hERG channel using 1 μ M Cas. We did not record any difference on hERG current before and after 1 μ M Cas perfusion. We have now included these new data in the revised version of the manuscript (Suppl. Fig. 6).

Q2. The splice variant of CaV1.3 used in the recombinant experiments is the exon 42a version, corresponding to a shortened form of the channel. It might be useful to confirm that the toxin also lacks the ability to block the long CaV1.3 isoform.

R2. We thank the reviewer for pointing out this issue. We performed new experiments using the long form of CaV1.3 (Cav1.3L) isoform. We did not observe any reduction of the related Ca current after 300nM and 1 μ M Cas. We have added this new data in the revised version (Fig. 3, Suppl. Fig. 3, Suppl. Fig. 4)

Reviewers' Comments:

Reviewer #1:

Remarks to the Author:

The authors have addressed all my initial concerns. This is overall a superb story and valuable resource for the community!

Reviewer #2:

Remarks to the Author:

I am satisfied with all revisions and responses to comments.